# How Did Breast Cancer Patients Fare during Different Phases of the COVID-19 Pandemic in Norway Compared to Age-Matched Controls?

**DOI:** 10.3390/cancers16030602

**Published:** 2024-01-31

**Authors:** Karianne Svendsen, Sigrid Leithe, Cassia B. Trewin-Nybråten, Aina Balto, Lise Solberg Nes, Anders Meland, Elin Børøsund, Cecilie E. Kiserud, Kristin Valborg Reinertsen, Hege R. Eriksen, Ylva Maria Gjelsvik, Giske Ursin

**Affiliations:** 1Cancer Registry of Norway, Norwegian Institute of Public Health, 0379 Oslo, Norway; sile@kreftregisteret.no (S.L.); catr@kreftregisteret.no (C.B.T.-N.); aiba@kreftregisteret.no (A.B.); ylgj@kreftregisteret.no (Y.M.G.); 2Department of Nutrition, Institute of Basic Medical Sciences, Faculty of Medicine, University of Oslo, 0317 Oslo, Norway; 3The Lipid Clinic, Department of Endocrinology, Morbid Obesity and Preventive Medicine, Oslo University Hospital, 0586 Oslo, Norway; 4Department of Digital Health Research, Division of Medicine, Oslo University Hospital, 0586 Oslo, Norway; lise.solberg.nes@rr-research.no (L.S.N.); elin.borosund@rr-research.no (E.B.); 5Department of Psychiatry and Psychology, Mayo Clinic College of Medicine and Science, Mayo Clinic, Rochester, MN 55905, USA; 6Institute of Clinical Medicine, Faculty of Medicine, University of Oslo, 0318 Oslo, Norway; 7Department of Social Sciences, Norwegian School of Sport Sciences, 0863 Oslo, Norway; andersme@nih.no; 8Department of Nursing and Health Sciences, Faculty of Health and Social Sciences, University of South-Eastern Norway, 3054 Drammen, Norway; 9Department of Oncology, Oslo University Hospital, 0310 Oslo, Norway; ckk@ous-hf.no (C.E.K.); kvr@ous-hf.no (K.V.R.); 10Department of Sport, Food and Natural Sciences, Western Norway University of Applied Sciences, 5020 Bergen, Norway; hrer@hvl.no; 11Department of Population and Public Health Sciences, University of Southern California, Los Angeles, CA 90033, USA

**Keywords:** COVID-19, HRQoL, breast cancer, EORTC, family

## Abstract

**Simple Summary:**

This study of Norwegian breast cancer cases and controls invited during different COVID-19 phases (social restrictions, high infection rate and post-COVID-19 phases) found consistently lower health-related quality of life (HRQoL) in women with breast cancer but minor differences across the three phases of the pandemic. Among breast cancer cases, young women who were living with children <18 years of age had the most HRQoL difficulties, whereas among controls, single women encountered the most difficulties. Living with children <18 had divergent effects on several HRQoL domains amongst cases and controls, with worse scores for breast cancer cases but better scores for controls. Hence, the burden of a cancer diagnosis (i.e., fatigue, worry, guilt, etc.) might be even greater among women with young children.

**Abstract:**

Little is known about how health-related quality of life (HRQoL) in breast cancer cases differed from that of controls during and after the COVID-19 pandemic. This study used data from an ongoing, nationwide HRQoL survey of 4279 newly diagnosed breast cancer cases and 2911 controls to investigate how breast cancer patients fared during different phases of the pandemic compared to controls. Responders during 2020–2022 were categorized into three COVID-19-related phases: the social restrictions phase, the high infection rate phase, and the post-pandemic phase. Across phases, breast cancer cases had significantly worse scores in most HRQoL domains compared to controls. Apart from slightly more insomnia in the high infection rate phase for both cases and controls, and better social functioning for young cases in the post-COVID-19 phase, the case-control differences in HRQoL remained consistent across phases. When the phases were assessed as one period, young women and those living with children <18 years of age fared the worst among breast cancer cases, while single women fared the worst among controls. In contrast, controls living with children <18 years of age exhibited better HRQoL than controls without children. In summary, women with breast cancer did not appear to fare differently than controls in terms of HRQoL across COVID-19 phases. However, breast cancer cases with young children fared worse in their HRQoL than other breast cancer cases.

## 1. Introduction

In March 2020, the World Health Organization (WHO) declared the COVID-19 outbreak a global pandemic [1]. The following two years were characterized by several waves of high infection rates and varying degrees of social restriction [2]. In Norway, the containment measures at the start of the pandemic were strong, resulting in low infection rates compared to other countries [3].

Worldwide, the pandemic severely impacted healthcare systems [4]. In Norway, the mammographic screening program was paused for several months during the spring of 2020, possibly resulting in diagnostic delays for breast cancer [5,6,7,8]. Postponed visits and disruptions in surgical procedures and treatment may have caused anxiety and concerns about cancer prognosis [9]. These circumstances could thus have exacerbated the already worse health-related quality of life (HRQoL) in women with breast cancer compared to apparently healthy controls [10,11,12,13,14]. HRQoL covers the subjective perceptions of overall quality of life, involving physical, emotional, social, and cognitive functions in addition to symptom scales including fatigue, pain, and insomnia [15]. Two studies reported that breast cancer cases had similar, or reduced, HRQoL during the early part of the pandemic compared to before [2,16]. Comparable results have also been noted in studies involving general populations [17,18]. Cross-sectional data from Norway suggest that periods of social restrictions may have affected youths, singles, and unemployed people the hardest [19]. Such social isolation may have resulted in mental distress and unhealthy lifestyle behaviors, including poor sleep [20]. However, whether the pandemic and post-pandemic period impacted HRQoL differently in women with breast cancer compared with controls, and whether other COVID-19 related factors impacted HRQoL, is unknown. From a public health perspective, it is important to document if HRQoL in women with breast cancer varied during different phases of the COVID-19 pandemic because it can provide valuable information on how vulnerable patient populations coped with self-isolation and uncertainty. This is in line with the WHO report “Imagining the Future of Pandemics and Epidemics” [21] that emphasizes the need to address and proactively monitor public health and social measures related to mental health to prepare populations for future infectious threats, natural disasters, and other crises [21].

This study aimed to describe self-reported HRQoL in women newly diagnosed with breast cancer and controls during three distinct phases of the COVID-19 pandemic. Further, the study aimed to investigate whether age, region of residence, relationship status, or living with children <18 years of age modified any differences in HRQoL between cases and controls.

## 2. Materials and Methods

This was a cross-sectional study conducted during 2020–2022 among women aged ≥ 18 years of age with incident breast cancer (*N* = 10,242) identified from the Cancer Registry of Norway (CRN) and controls (*N* = 11,364) identified from the general Norwegian population. The CRN has collected population-based patient-reported outcomes from breast cancer cases and controls in the ongoing CRN HRQoL survey since September 2020 [22]. The survey is also sent to cases and controls with other cancer types.

### 2.1. Cases and Controls

Information on breast cancer diagnoses (ductal carcinoma in situ or invasive breast cancer) including age at diagnosis and other medical data related to the breast cancer diagnosis were obtained from the CRN [5]. Controls were selected randomly and frequency-matched to the expected distribution of incident breast cancer cases (based on the previous five years’ breast cancer incidence) across ten-year age groups and region of residence.

Users of the official digital health communication platform or digital mailbox in Norway were eligible for invitation to the CRN HRQoL survey. This encompassed about 84% (*n* = 8710) of incident breast cancer cases diagnosed between 2020 and 2022, and 78% (*n* = 9004) of corresponding controls [21].

### 2.2. The CRN HRQoL Survey

This CRN HRQoL survey included validated questionnaires developed by the European Organization for Research and Treatment of Cancer (EORTC), including the EORTC QLQ-C30 questionnaire that measures global HRQoL, functions, and symptoms of HRQoL [23]. The survey furthermore included self-reported background data on age, relationship status (single or in a relationship/in a cohabitating relationship), information on cohabitating children under the age of 18 (yes or no), educational levels (primary or secondary school, college/university ≤4 years or >4 years), smoking habits (never, former, current smoker), body mass index in kg/m^2^ (grouped according to being normal weight, overweight, and obese), physical activity level (no exercise but physically active ≤ 3 hours/week (h/w), no exercise but physically active > 3 h/w, exercise 0–1 h/w, exercise 2–3 h/w and exercise 4+ h/w), and alcohol consumption (do you consume alcohol, yes or no). Missing data for each category were also presented.

### 2.3. Study Sample According to COVID-19 Phases

This study included cases and controls who received the initial, digital CRN HRQoL survey between September 2020 and December 2022. During this period, the goal was to invite cases to the survey around (but not before) 21 days after breast cancer diagnosis. The survey invitation timing was selected to ensure that cases were not invited to participate prior to receiving information about their diagnosis from their physicians [22]. The survey was sent on a similar date to controls, with a 30-day response period for both groups.

The average response rate in the study was 49% (*n* = 4279) for cases and 31% (*n* = 2911) for controls. Responders were categorized into one of three COVID-19 phases according to month of survey invitation:*Phase 1—“lockdown” (September 2020–September 2021):* Phase 1 was characterized by extended periods of lockdown or social restrictions until Norway officially reopened on 25 September 2021 [24]. The mammographic screening program was not closed during this period, but some places may have operated at a slower pace. Number of responders during phase 1: *n* = 1818 cases and 1690 controls.*Phase 2—“high infection rates” (October 2021–February 2022):* Phase 2 was characterized by few social restrictions, but the Omicron variant resulted in a peak in COVID-19 infections in Norway [25]. Number of responders during phase 2: *n* = 886 cases and 318 controls.*Phase 3—“post-pandemic” (March 2022–December 2022):* Phase 3 was characterized by “back to normal” as Norway ceased testing and monitoring of new COVID-19 cases [24]. Number of responders during phase 3: *n* = 1585 cases and 903 controls.

### 2.4. Statistical Analysis

Categorical variables were described by frequencies and proportions, and continuous variables were described by means and standard deviations (SD). Differences in sociodemographic and lifestyle variables between cases and controls were assessed using chi-square tests and Student’s *t*-tests and are presented for all phases combined and separately in Section 3. All single items within the EORTC QLQ-C30 questionnaire, global HRQoL, and functioning and symptom scores were transformed to a 0 to 100 scale according to the EORTC scoring manual [23]. Only participants who had answered all items within a domain received a score for that domain (for example, social functioning). Higher scores for global HRQoL and functional scales implied more functioning, and lower scores on symptom scales implied less symptoms and were thus considered beneficial [26].

Multivariable linear regression models were used to compare global HRQoL, functional scales, and symptom scales within and between cases and controls. Case–control differences were presented with 95% confidence intervals (95% CI). Statistical significance was assigned when the 95% CI did not include zero. The models included adjustments for potential confounders: 5-year age group, self-reported body mass index, educational level, smoking habits, physical activity level, and alcohol consumption (yes/no) (all variables grouped as displayed in Section 3). Case–control differences in HRQoL domains across COVID-19 phases obtained from the multivariable models are presented in figures (Section 3.1). A likelihood ratio test was used to assess the significance of potential interactions between case–control differences in HRQoL and the COVID-19 phases.

A likelihood ratio test was used to assess potential interactions between HRQoL scores, subgroups of age, young (18–49 years) versus older (50+ years) participants, region of residence (capital region (Oslo/Viken) versus rest of the country), relationship status (single versus in a relationship), and living with children <18 years of age (yes/no) and the three COVID-19 phases. These factors were considered to potentially have had an impact on HRQoL during the pandemic. For instance, both social restrictions and infection rates had more profound impacts in the capital region of Norway compared to rest of the country [3]. Case–control differences in HRQoL domains between subgroups in all phases combined were assessed using multivariable linear regression and are presented in figures (Section 3.1) and Appendix A.

Due to the exploratory nature of this study, no correction for multiple testing was performed. Between-group differences were, when applicable, evaluated against the minimally important difference for each HRQoL domain according to estimates for advanced breast cancer [27].

All analyses were performed in Stata version 18.0 (StataCorp 2021, Stata Statistical Software: Release 18; StataCorp LLC., College Station, TX, USA).

## 3. Results

Mean age was 58 ± 11 years at breast cancer diagnosis for cases and 59 ± 11 years at survey completion for controls (Table 1). Over 90% of cases responded to the survey within three months after diagnosis. Regarding treatment, 29% responded before surgery, whereas 56% had breast-conserving therapy before survey response (Table 2). Most cases and controls (77%) were in a relationship, and 23% of cases and 20% of controls were living with children <18 years of age. In total, 26% of both cases (*n* = 747) and controls (*n* = 1097) were residing in the capital area of Norway (Oslo and Viken) (Table 1).

Out of all the participants, most cases (42%) and controls (58%) participated in phase 1 (hence, they were invited to the survey in phase 1). Phase 2 included 21% of cases and 11% of controls, while phase 3 comprised 37% of cases and 31% of controls. Most cases had stage I breast cancer (42% in the lockdown phase, 47% in the high infection rate phase, and 40% in the post-pandemic phase) (Table 2). The proportion of young (18–49 years) and older (50+) women, the proportion in a relationship, and the proportion of those living with children <18 years of age were similar for responders from the three phases of the pandemic. There were, however, slightly fewer single, obese (BMI ≥30) control women and a higher proportion of physically inactive (no exercise but physically active ≤ 3 h/w) cases in the high infection rate phase compared to in the other two phases (Table 3). 

### 3.1. HRQoL across COVID-19 Phases

Overall, cases consistently reported worse scores on global HRQoL and all HRQoL functioning scales and worse scores on fatigue, nausea, pain, appetite loss, constipation, and financial difficulties than controls throughout the three phases of the COVID-19 pandemic (Figure 1 and Figure 2). The largest case–control difference was evident for role functioning, ranging from −19.6 (−21.4, −17.8) in phase 1 to −18.5 (−22.0, −14.9) in phase 2 and −18.0 (−20.3, −15.8) in phase 3. Conversely, the smallest (non-statistically significant) case–control differences were observed for dyspnea (0.35 (−1.4, 2.1) in phase 3] and diarrhea (−0.46 (−2.0, 1.1) in phase 1) (Figure 1 and Figure 2).

Symptom scores varied between the COVID phases for insomnia (cases: *p* = 0.02; controls: *p* = 0.06), diarrhea (cases: *p* = 0.03), financial difficulties (cases: *p* = 0.05), and dyspnea (controls: *p* = 0.01). The insomnia score differed the most between cases and controls in phase 1 (6.6 (4.6, 8.5)] (Figure 2) due to a high score in cases [33.8 (32.5, 35.2)] and a low score in controls [27.3 (25.9, 28.6)]. However, cases and controls in phase 2 reported the highest scores for insomnia [34.3 (32.4, 36.3)] for cases and 31.1 (27.9, 34.3) for controls.

No functioning scores varied significantly between the COVID phases for the overall population (Figure 1). However, the case–control difference in social functioning differed across phases in the young age group (*p* = 0.04). This was due to the overall lowest score for controls in phase 3 but marginally better scores for young breast cancer cases in terms of social functioning in the post-pandemic phase 3 (62.3 ± 1.3) compared to the score in the preceding phases (59.4 ± 1.2 and 56.4 ± 1.8 in phase 1 and 2, respectively). Minor variation in the size of the case–control differences across the three COVID phases was otherwise detected.

### 3.2. COVID19-Related Factors

When the COVID-19 phases were assessed as one period, young women had worse scores for HRQoL functioning and fatigue compared to older women among both cases and controls (Figure 3 and Figure 4).

In particular, there were significant differences between young breast cancer cases and young controls in global **HRQoL** (−12.1 (−14.2, 9.9)], social functioning (−23.5 (−26.0, −20.9)], role functioning (−27.3 (−30.2, −24.4)], and fatigue [14.8 (12.4, 17.3)] (Figure 3 and Figure 4). The magnitude of the differences between cases and controls living with children <18 years of age was similar as for the young age group for global HRQoL, social functioning, role functioning, and fatigue. All these case–control differences were due to worse scores in cases and better scores in controls among those living with children <18 years of age compared to those not living with children <18 years of age. For cases and controls in a relationship, an observed large case–control difference in social functioning (−15.6 (−16.9, −14.3)] was due to a low score in cases and a high score in controls. On the other hand, there were minor differences (score difference <3) for physical and cognitive functioning, dyspnea, insomnia, and pain between single cases and controls (Figure 3 and Figure 4).

There was no significant case–control difference in HRQoL between women living in the capital region of Norway compared to the rest of the country (Appendix A).

## 4. Discussion

Breast cancer cases reported significantly worse scores on all HRQoL functioning scales and most HRQoL symptom scales, including fatigue and pain, compared with controls in all phases. The case–control differences in global HRQoL, social and role functioning, and fatigue were greater than what has been defined as minimally important differences for advanced breast cancer [27]. There were, however, some minor variations in the case–control differences across phases. Cases experienced most insomnia during social restriction and high infection rate phases, whereas controls experienced most insomnia during the high infection rate phase. The problems with insomnia aligned with the results of a study of the Italian general population that found that one of the main consequences of social restrictions was poor sleep [21]. Young women with breast cancer had significantly better social functioning in the post-COVID-19 phase compared to the preceding phases. However, the size of the case–control difference across phases was less than what was considered to be a minimally important difference [27] for both insomnia and social functioning. Hence, the pandemic seemed to have negligible impact on HRQoL in women with and without breast cancer in Norway. For women with breast cancer this could partly be because cancer diagnosis may have taken precedence in this situation. For both groups, a possible explanation could be that Norway felt a milder impact from the pandemic in phase 2 in terms of infection and death rates compared to other European countries [3]. This was due to rapid and strong containment measures [3]. Still, a recent review on the general population incorporating data across continents concluded that people were surprisingly resilient over time and recovered quickly from COVID-19 restrictions [28]. In line with these findings in breast cancer patients, a study from the Netherlands [2] found minor differences in HRQoL measures during the early phase of the pandemic (compared to the pre-pandemic period). Another study from Spain [29] found that breast cancer patients seemed to adapt well to the COVID-19 pandemic. On the contrary, a single-center study from the USA showed that HRQoL in breast cancer patients was worsened by the COVID-pandemic (compared to the pre-pandemic period) [30]. The current study expands on these findings by comparing breast cancer cases to controls and by showing that, in general, the magnitude of the case–control differences was similar across phases and also in the post-COVID-19 phase.

Independent of the COVID-19 pandemic, breast cancer patients who were young and/or living with children <18 years of age fared the worst in terms of HRQoL. Moreover, living with children <18 years of age had divergent associations for several HRQoL domains for cases versus controls. For cases, living with children <18 years of age was associated with particularly poor scores for global HRQoL, social and role functioning, and fatigue, whereas controls exhibited better scores compared to groups without cohabitating children. This resulted in larger, above minimally important differences in these HRQoL domains between cases and controls with cohabitating children <18 years of age [26]. The lower global HRQoL and social and role functioning scores in cases living with children could potentially be due to guilt, sadness, and depressive symptoms if the cancer inhibited them from actively participating in their children’s lives [31]. Since most children <18 years of age are enrolled in school or pre-school, we could speculate that the lower HRQoL in breast cancer cases could be due to the extra burden of homeschooling during the pandemic, and numerous social events after the pandemic. In any case, it would pose a challenge for women with breast cancer who find it difficult to actively participate in their children’s life. Hence, the added challenge of raising children can be even harder for women with breast cancer, resulting in worse HRQoL in this group, including fatigue. On the other hand, family plays a crucial role as the primary source of social support for cancer patients, significantly influencing coping mechanisms and overall well-being in a positive way [32].

We found that social functioning scores differed more between cases and controls who were in a relationship than between single cases and controls. This could be attributed to the numerous social restrictions associated with a cancer diagnosis that may constitute a larger burden when you have a partner. However, the effect of social support on quality of life is also affected by other factors [33]. The same divergent case–control pattern was observed for relationship status (e.g., that having a partner was negative for controls but positive for cases); however, the effect sizes were smaller and less consistent than for the group with young children.

Breast cancer affects a large proportion of women globally. It is thus important to find tools to provide support to ease the stress that comes with a cancer diagnosis [34] so that women with breast cancer can focus their energy on things that matter to them, such as family life.

Conversely, it is noteworthy that, throughout the study period, single cancer-free controls demonstrated comparable scores to single breast cancer cases across numerous HRQoL domains. These results therefore expand on short-term data from the first COVID-19 phase in the Norwegian population that showed that young individuals and singles fared the worst during the COVID-19 pandemic [19]. This result may have a large impact on the general population as Norwegian statistics show that about 20% of the population live alone [35]. While it has previously been proposed that the mental health of singles declined during the pandemic [2], our findings, indicating similar HRQoL in single women with breast cancer and controls (without breast cancer) both during and after the pandemic, underscore the significant and potential long-term consequences of this issue. The same short-term data from the first COVID-19 phase in Norway found that those residing in larger cities (with the strongest social restrictions) fared the worst [19]. In our study, no distinctions in HRQoL were identified when comparing cases and controls in the capital region with all other counties combined.

Overall, the findings of the current study provide valuable insights into specific and large patient subgroups that warrant close monitoring in the light of future events [21].

### Strengths and Limitations

The strengths of the study include the large sample size, the cancer-free control group, and data across three phases of the COVID-19 pandemic. This is in contrast to other studies of HRQoL in women with breast cancer during the pandemic which lacked a control group and post-pandemic comparison group [2,16,28,29,30]. As mandated by law, reporting of cancer cases to the CRN ensures a complete dataset of all diagnosed breast cancer patients [36]. All breast cancer patients above 18 years of age with digital access (84%) were available for inclusion. Moreover, the study was nationwide. However, the moderate response rates (49% of breast cancer cases and 31% of controls who could be reached digitally) may have led to some selection bias. Our findings for HRQoL overall were, however, in line with previous studies. Although we think it is unlikely, we cannot exclude the possibility that the magnitude of this bias changed throughout the phases of the pandemic. Another limitation is that there were fewer controls in phase 2 compared to phases 1 and 3. Some lifestyle characteristics suggest that cases and controls belonging to this phase were slightly different than in the other two phases, which could have impacted the results. However, similar patterns were observed when comparing results in phase 1 to phase 2 and 3 combined. Finally, it should also be noted that the EORTC QLQ-C30 questionnaire aims to assess HRQoL independent of the pandemic, and we cannot rule out that factors other than the COVID-19 pandemic may have impacted the HRQoL scores in cases and controls during different phases of the pandemic.

## 5. Conclusions

In conclusion, the sizes of the HRQoL case–control differences were similar across COVID-19 phases. Hence, women with breast cancer appeared to be remarkably resilient during the COVID-19 pandemic, despite their disease and all additional concerns that came with the pandemic. The results also show that in terms of HRQoL, independent of the pandemic, young breast cancer cases living with children <18 years of age fared worst among women with breast cancer, while single women fared worst among the controls. Hence, the burden (i.e., load, fatigue, worry, bad consciousness, guilt, etc.) of a cancer diagnosis and experience might be exacerbated by living with young children, which may also be the case for some domains for those having a partner. Consequently, follow-ups with women with breast cancer should be adapted according to the high-risk groups identified in this study. The findings in this study highlight the need for preventive measures targeting specific patient (and non-patient) groups in the event of a future pandemic.

## Figures and Tables

**Figure 1 cancers-16-00602-f001:**
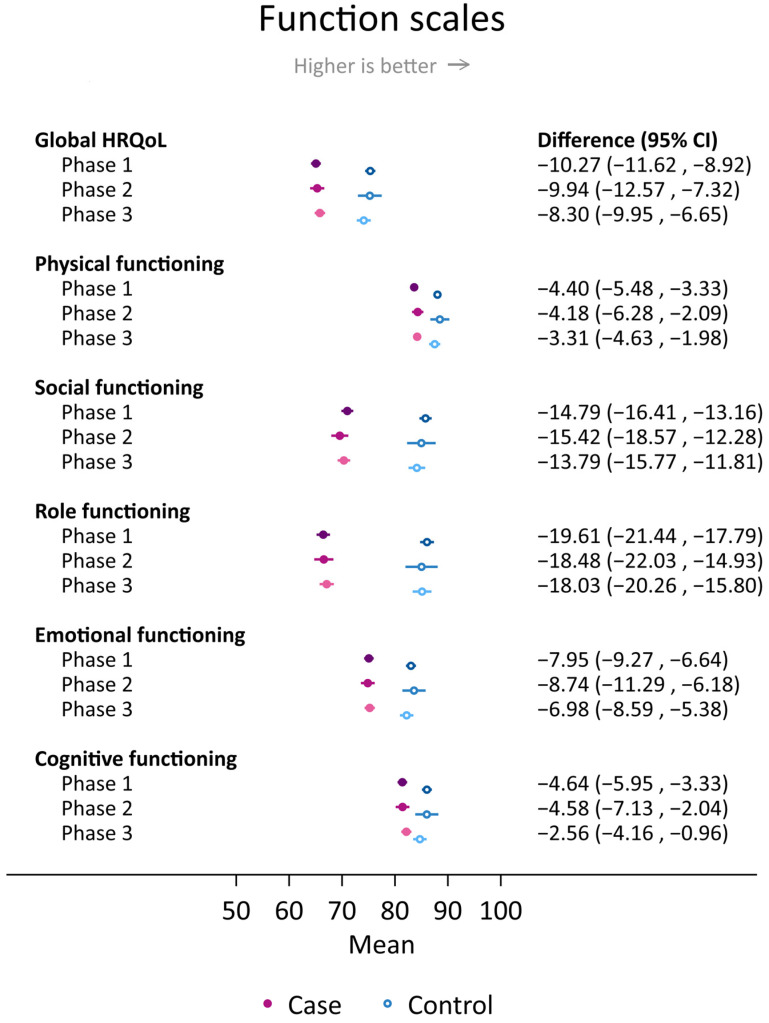
Mean scores and differences between cases and controls with 95% confidence intervals in global health-related quality of life (HRQoL) and HRQoL functioning scales among cases and controls who responded to the Cancer Registry of Norway HRQoL survey during COVID phases: phase 1 “lockdown”, September 2020–September 2021; phase 2 “high infection rates”; October 2021-February 2022; “phase 3 post-pandemic”, March 2022–December 2022. Results were adjusted for 5-year age group, body mass index group, smoking, educational level, physical activity level, and alcohol consumption (yes/no). Different shades of purple (cases) and blue (controls) are used to visually distinguish between scores in phases 1, 2 and 3.

**Figure 2 cancers-16-00602-f002:**
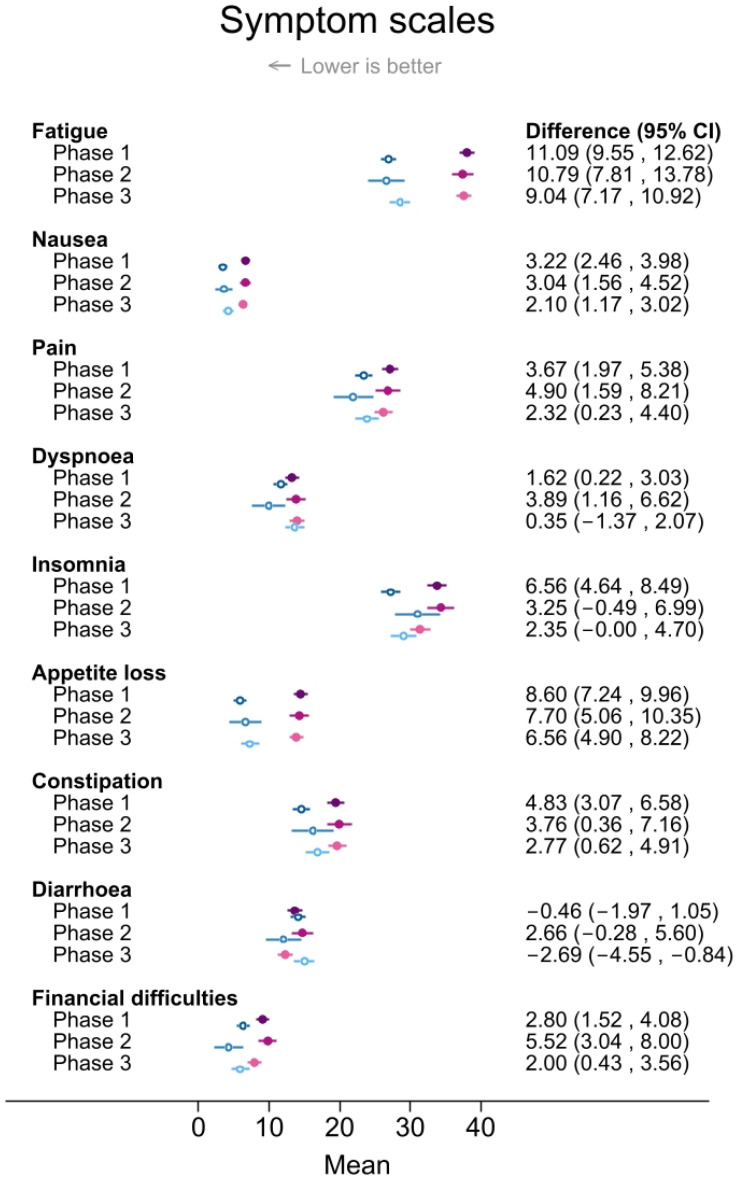
Mean scores and differences between cases and controls with 95% confidence intervals in health-related quality of life (HRQoL) symptom scales among cases and controls who responded to the Cancer Registry of Norway HRQoL survey during COVID phases: phase 1 “lockdown”, September 2020–September 2021; phase 2 “high infection rates”, October 2021–February 2022; “phase 3 post-pandemic”, March 2022–December 2022. Results were adjusted for 5-year age group, body mass index group, smoking, educational level, physical activity level, and alcohol consumption (yes/no). Different shades of purple (cases) and blue (controls) are used to visually distinguish between scores in phases 1, 2 and 3.

**Figure 3 cancers-16-00602-f003:**
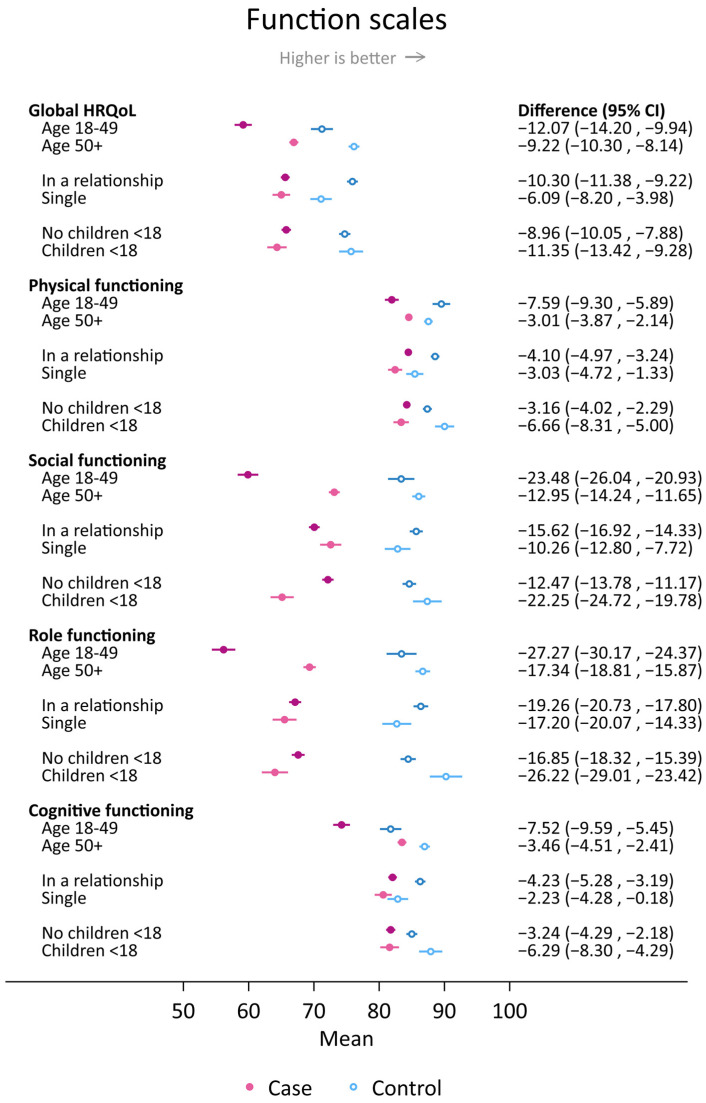
Mean scores and differences between cases and controls with 95% confidence intervals in selected measures of global health-related quality of life (HRQoL) and HRQoL functioning scales during September 2020–December 2022 stratified by age group, relationship status, and whether or not respondents lived with children <18 years of age. Results were adjusted for 5-year age group (except for when age groups were compared), body mass index group, smoking, educational level, physical activity level, and alcohol consumption (yes/no).

**Figure 4 cancers-16-00602-f004:**
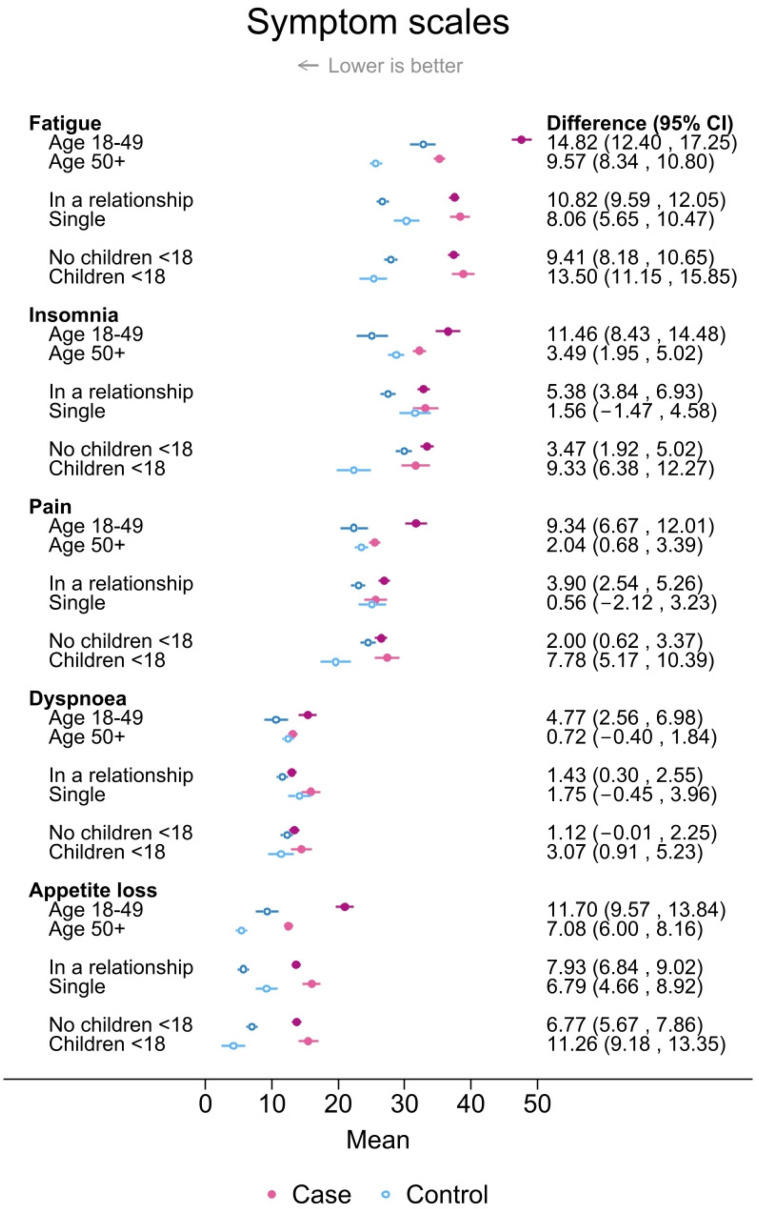
Mean scores and differences between cases and controls with 95% confidence intervals in selected measures of health-related quality of life (HRQoL) symptom scales during September 2020–December 2022 stratified by age group, relationship status, and whether or not respondents lived with children <18 years of age. Results were adjusted for 5-year age group (except for when age groups were compared), body mass index group, smoking, educational level, physical activity level, and alcohol consumption (yes/no).

**Table 1 cancers-16-00602-t001:** Sociodemographic characteristics of breast cancer cases and controls.

Variables	Case (*N* = 4279)	Control (*N* = 2911)	*p*
**Age**, mean (SD)	58.0 (11.4)	59.1 (11.2)	<0.01
**BMI**, mean (SD)	26.1 (5.1)	25.8 (4.8)	0.01
**Residing in capital area**, *n* (%)	747 (25.7)	1097 (25.6)	
**Relationship status**, *n* (%)			0.99
In a relationship	3288 (76.8)	2232 (76.7)	
Not in a relationship	847 (19.4)	575 (19.8)	
Missing	144 (3.4)	104 (3.6)	
**Children <18 living at home**, *n* (%)			<0.01
No	3153 (73.7)	2248 (77.2)	
Yes	993 (23.2)	568 (19.5)	
Missing	133 (3.1)	95 (3.3)	
**Employed**, *n* (%)			<0.01
No	637 (14.9)	516 (17.7)	
Yes	2507 (58.6)	1710 (58.7)	
No, retired	936 (21.9)	542 (18.6)	
Missing	199 (4.7)	143 (4.9)	
**Educational level**, *n* (%)			0.03
Primary school	305 (7.1)	238 (8.2)	
Secondary school	1503 (35.1)	1037 (35.6)	
College/university ≤ 4 years	1255 (29.3)	890 (30.6)	
College/university > 4 years	1079 (25.2)	655 (22.5)	
Missing	137 (3.2)	91 (3.1)	
**Exercise levels**, *n* (%)			<0.01
No exercise. Physically active ≤ 3 h/w	564 (13.2)	323 (11.1)	
No exercise. Physically active > 3 h/w	1423 (33.3)	796 (27.3)	
Exercise 0–1 h/w	1044 (24.4)	764 (26.2)	
Exercise 2–3 h/w	815 (19.0)	677 (23.3)	
Exercise 4+ h/w	279 (6.5)	253 (8.7)	
Missing	154 (3.6)	98 (3.4)	
**Smoking**, *n* (%)			<0.01
Never smoker	1930 (45.1)	1366 (46.9)	
Former smoker	1809 (42.3)	1114 (38.3)	
Current smoker	408 (9.5)	334 (11.5)	
Missing	132 (3.1)	97 (3.3)	
**Alcohol consumption**, *n* (%)			<0.01
No	1166 (27.2)	683 (23.5)	
Yes	2977 (69.6)	2133 (73.3)	
Missing	136 (3.2)	95 (3.3)	
**BMI group (kg/m^2^)**, *n* (%)			0.19
<25	1779 (41.6)	1260 (43.3)	
25–29	1454 (34.0)	974 (33.5)	
≥30	865 (20.2)	545 (18.7)	
Missing	181 (4.2)	132 (4.5)	

Data on cases and controls who responded to the Cancer Registry of Norway health-related quality of life survey during the COVID-19 phases: “lockdown”, September 2020–September 2021; “high infection rates”, October 2021–February 2022; and “post-pandemic”, March 2022–December 2022. *p*-values are obtained using a chi-square test for categorical variables and a Student’s *t*-test for continuous variables.

**Table 2 cancers-16-00602-t002:** Medical data on breast cancer cases across COVID-19 phases.

Medical Data	Total *N* = 4279	Phase 1 “*Lockdown*” *N* = 1818	Phase 2 *“High Infection Rate*” *N* = 886	Phase 3 *“Post-Pandemic*” *N* = 1575
**Age**, mean (SD)	58.0 (11.4)	57.7 (11.4)	58.3 (11.1)	58.2 (11.6)
**Stage**, *n* (%)				
I	1815 (42.4)	766 (42.1)	420 (47.4)	629 (39.9)
II	1069 (25.0)	462 (25.4)	195 (22.0)	412 (26.2)
III	312 (7.3)	134 (7.4)	60 (6.8)	118 (7.5)
IV	84 (2.0)	40 (2.2)	15 (1.7)	29 (1.8)
Unknown	999 (23.3)	416 (22.9)	196 (22.1)	387 (24.6)
**HER2 status**, *n* (%)				
Negative	3226 (75.4)	1354 (74.5)	691 (78.0)	1181 (75.0)
Positive	442 (10.3)	194 (10.7)	79 (8.9)	169 (10.7)
Unknown	611 (14.3)	270 (14.9)	116 (13.1)	225 (14.3)
**ER status**, *n* (%)				
Negative	480 (11.2)	198 (10.9)	98 (11.1)	184 (11.7)
Positive	3242 (75.8)	1372 (75.5)	681 (76.9)	1189 (75.5)
Unknown	557 (13.0)	248 (13.6)	107 (12.1)	202 (12.8)
**PR status**, *n* (%)				
Negative	1063 (24.8)	446 (24.5)	209 (23.6)	408 (25.9)
Positive	2606 (60.9)	1101 (60.6)	561 (63.3)	944 (59.9)
Unknown	610 (14.3)	271 (14.9)	116 (13.1)	223 (14.2)
**Treatment status at survey response**, *n* (%)				
Responded prior to surgery	1240 (29.0)	482 (26.5)	244 (27.5)	514 (32.6)
BCT	2396 (56.0)	1033 (56.8)	500 (56.4)	863 (54.8)
BCT + radiation therapy	226 (5.3)	130 (7.2)	52 (5.9)	44 (2.8)
Mastectomy	400 (9.3)	163 (9.0)	88 (9.9)	149 (9.5)
Mastectomy + radiation therapy	7 (0.2)	7 (0.4)	0	0
Radiation therapy, surgery unknown	10 (0.2)	3 (0.2)	2 (0.2)	5 (0.3)
**Time from diagnosis to response**, *n* (%)				
≤1 month	1938 (45.3)	797 (43.8)	370 (41.8)	771 (49.0)
2–3 months	2106 (49.2)	897 (49.3)	469 (52.9)	740 (47.0)
4–6 months	65 (1.5)	29 (1.6)	21 (2.4)	15 (1.0)
>6 months	8 (0.2)	4 (0.2)	2 (0.2)	2 (0.1)
Unknown	162 (3.8)	91 (5.0)	24 (2.7)	47 (3.0)

HER2: Human Epidermal Growth Factor Receptor 2; ER: estrogen receptor; PR: progesterone receptor; BCT: breast-conserving therapy.

**Table 3 cancers-16-00602-t003:** Sociodemographic data on breast cancer cases and controls across COVID-19 phases.

	Phase 1 “*Lockdown*”			Phase 2 “*High Infection Rate*”			Phase 3 “*Post-Pandemic*”		
Variables	Case (*N* = 1818)	Control (*N* = 1690)	*p*	Case (*N* = 886)	Control (*N* = 318)	*p*	Case (*N* = 1575)	Control (*N* = 903)	*p*
**Age**, mean (SD)	57.7 (11.4)	58.9 (11.0)	<0.01	58.3 (11.1)	59.6 (11.0)	0.07	58.2 (11.6)	59.4 (11.8)	0.02
**BMI**, mean (SD)	26.3 (5.2)	25.7 (4.8)	<0.01	26.0 (4.9)	25.5 (4.5)	0.11	25.9 (5.0)	25.9 (4.8)	0.97
**Relationship status**, *n* (%)			0.70			0.43			0.86
In a cohabitating relationship	1290 (71.0)	1184 (70.1)		611 (69.0)	222 (69.8)		1137 (72.2)	657 (72.8)	
In a relationship, but not living together	105 (5.8)	97 (5.7)		58 (6.5)	24 (7.5)		87 (5.5)	48 (5.3)	
Not in a relationship	353 (19.4)	348 (20.6)		174 (19.6)	52 (16.4)		320 (20.3)	175 (19.4)	
Missing	70 (3.9)	61 (3.6)		43 (4.9)	20 (6.3)		31 (2.0)	23 (2.5)	
**Children <18 living at home**, *n* (%)			0.00			0.84			0.02
No	1311 (72.1)	1302 (77.0)		652 (73.6)	231 (72.6)		1190 (75.6)	715 (79.2)	
Yes	436 (24.0)	326 (19.3)		194 (21.9)	71 (22.3)		363 (23.0)	171 (18.9)	
Missing	71 (3.9)	62 (3.7)		40 (4.5)	16 (5.0)		22 (1.4)	17 (1.9)	
**Educational level**, *n* (%)			0.40			0.06			0.29
Primary school	127 (7.0)	124 (7.3)		56 (6.3)	32 (10.1)		122 (7.7)	82 (9.1)	
Secondary school	649 (35.7)	610 (36.1)		314 (35.4)	102 (32.1)		540 (34.3)	325 (36.0)	
College/university ≤ 4 years	519 (28.5)	512 (30.3)		244 (27.5)	98 (30.8)		492 (31.2)	280 (31.0)	
College/university > 4 years	454 (25.0)	384 (22.7)		230 (26.0)	71 (22.3)		395 (25.1)	200 (22.1)	
Missing	69 (3.8)	60 (3.6)		42 (4.7)	15 (4.7)		26 (1.7)	16 (1.8)	
**Employed**, *n* (%)			0.00			0.26			0.36
No	349 (19.2)	381 (22.5)		105 (11.9)	27 (8.5)		183 (11.6)	108 (12.0)	
Yes	1047 (57.6)	1007 (59.6)		516 (58.2)	188 (59.1)		944 (59.9)	515 (57.0)	
No, retired	302 (16.6)	199 (11.8)		219 (24.7)	84 (26.4)		415 (26.3)	259 (28.7)	
Missing	120 (6.6)	103 (6.1)		46 (5.2)	19 (6.0)		33 (2.1)	21 (2.3)	
**Exercise levels**, *n* (%)			0.00			0.00			0.00
No exercise Physically active ≤ 3 h/w	214 (11.8)	197 (11.7)		135 (15.2)	30 (9.4)		215 (13.7)	96 (10.6)	
No exercise Physically active > 3 h/w	622 (34.2)	472 (27.9)		278 (31.4)	81 (25.5)		523 (33.2)	243 (26.9)	
Exercise 0–1 h/w	424 (23.3)	442 (26.2)		220 (24.8)	91 (28.6)		400 (25.4)	231 (25.6)	
Exercise 2–3 h/w	344 (18.9)	383 (22.7)		149 (16.8)	72 (22.6)		322 (20.4)	222 (24.6)	
Exercise 4+ h/w	141 (7.8)	133 (7.9)		53 (6.0)	28 (8.8)		85 (5.4)	92 (10.2)	
Missing	73 (4.0)	63 (3.7)		51 (5.8)	16 (5.0)		30 (1.9)	19 (2.1)	
**Smoking**, *n* (%)			0.00			0.56			0.01
Never smoker	850 (46.8)	779 (46.1)		388 (43.8)	148 (46.5)		692 (43.9)	439 (48.6)	
Former smoker	748 (41.1)	643 (38.0)		352 (39.7)	122 (38.4)		709 (45.0)	349 (38.6)	
Current smoker	153 (8.4)	201 (11.9)		105 (11.9)	32 (10.1)		150 (9.5)	101 (11.2)	
Missing	67 (3.7)	67 (4.0)		41 (4.6)	16 (5.0)		24 (1.5)	14 (1.6)	
**Alcohol consumption**, *n* (%)			0.01			0.63			0.00
No	495 (27.2)	400 (23.7)		230 (26.0)	78 (24.5)		441 (28.0)	205 (22.7)	
Yes	1253 (68.9)	1230 (72.8)		616 (69.5)	225 (70.8)		1108 (70.3)	678 (75.1)	
Missing	70 (3.9)	60 (3.6)		40 (4.5)	15 (4.7)		26 (1.7)	20 (2.2)	
**BMI group (kg/m^2^)**, *n* (%)			0.04			0.25			0.69
<25	740 (40.7)	742 (43.9)		348 (39.3)	137 (43.1)		691 (43.9)	381 (42.2)	
25–29	591 (32.5)	558 (33.0)		316 (35.7)	108 (34.0)		547 (34.7)	308 (34.1)	
≥30	396 (21.8)	315 (18.6)		173 (19.5)	50 (15.7)		296 (18.8)	180 (19.9)	
Missing	91 (5.0)	75 (4.4)		49 (5.5)	23 (7.2)		41 (2.6)	34 (3.8)	

Data on cases and controls who responded to the Cancer Registry of Norway health-related quality of life survey during the COVID-19 phases: “lockdown”, September 2020–September 2021; “high infection rates”, October 2021–February 2022; and “post-pandemic”, March 2022–December 2022. *p*-values were obtained using a chi-square test for categorical variables and a Student’s *t*-test for continuous variables.

## Data Availability

Data available on request due to ethical restrictions, privacy laws in Norway, and the GDPR. The data presented in this study are available on request from the corresponding author, assuming all ethical and legal permissions are in place. The data are not publicly available due to ethical restrictions, privacy laws in Norway, and the GDPR.

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
