# Peer review of "How Did Breast Cancer Patients Fare during Different Phases of the COVID-19 Pandemic in Norway Compared to Age-Matched Controls?"

_cancers, 2024, doi:10.3390/cancers16030602_

Round 1

Reviewer 1 Report

Comments and Suggestions for Authors

This prospective study evaluates the psychological impact on breast cancer survivors and aged matched controls of the COVID-19 pandemic. the study is well conducted , comprehensive and has an impressive response rate by the target cohort -it is unclear from the text if this was a COVID-19 specific study or one that was developed and then conducted during the pandemic. The title should reflect that the study was conducted in Norway. The authors highlight the vulnerability of patients with breast cancer who are rearing children and emphasise appropriately the need for increased supports for this group. Breast cancer is the commonest cause of death for women rearing children globally - while the pandemic has gone disruptive events such as it haven't and persist include conflict, climate change related events and a tripling of the possibility of future pandemics in the decades ahead - it would be worthwhile highlighting this in the discussion - the importance of this study relate to its implications for future events I feel.

The results are presented in 2 figures - these figures are cluttered I would suggest splitting them into 4 or modifying them from landscape form - the figures are the disseminatable aspect of the study as such I feel they should be viewed from the perspective of how they would look on social media or in a powerpoint presentation 

the discussion is well written but paragraph 1 belongs further down in the discussion and seems out of context as a starting point for this section

many references eg 11,12,13,14,16..... are incomplete 

Comments on the Quality of English Language

minor grammatical change occasionally in text 

Reviewer 2 Report

Comments and Suggestions for Authors

The article titled " How did breast cancer patients fare during different phases of the COVID-19 pandemic compared to age-matched controls?" by Karianne Svendsen et al. offers valuable insights. However, there are several issues that require attention:

1. The introduction lacks a clear explanation of why it is important to study the impact of the COVID-19 pandemic on health-related quality of life (HRQoL) in breast cancer cases compared to controls. Providing a rationale or background information would help to contextualize the study.

2. The sample size of the study is mentioned, but there is no information provided on the selection criteria or representativeness of the participants. It would be beneficial to provide details on how the study participants were recruited to ensure the validity and generalizability of the findings.

3. The statement that breast cancer cases had significantly worse HRQoL than controls is made without providing any specific measures used to assess HRQoL. It is important to mention the specific domains or scales used to measure HRQoL and provide relevant statistical results to support the claim.

4. While it is mentioned that there were slight differences in insomnia and social functioning between cases and controls during specific phases, it would be helpful to provide more details on the magnitude of these differences and whether they were statistically significant.

5. The statement that breast cancer cases with young children fared worse than other breast cancer cases is interesting but lacks additional information on the potential reasons for this difference.  

By addressing these suggestions and providing additional information, the text will become more informative and robust, improving the overall clarity and validity of the study's findings.

Comments on the Quality of English Language

 Moderate editing of English language required

Round 2

Reviewer 2 Report

Comments and Suggestions for Authors The authors have addressed all of the listed reviewer comments, and the present form of the manuscript is accepted for publication.

Comments on the Quality of English Language

Minor editing of English language required